# Unusual Duodenal Ulcer: Metastatic Urothelial Carcinoma of the Renal Pelvis

**DOI:** 10.3390/diagnostics13142455

**Published:** 2023-07-24

**Authors:** Yoo Dong Won, Su Lim Lee, Kyung Jin Seo

**Affiliations:** 1Department of Radiology, Uijeongbu St. Mary’s Hospital, College of Medicine, The Catholic University of Korea, Seoul 06591, Republic of Korea; yoodong1@catholic.ac.kr (Y.D.W.); radlsl@catholic.ac.kr (S.L.L.); 2Department of Hospital Pathology, Uijeongbu St. Mary’s Hospital, College of Medicine, The Catholic University of Korea, Seoul 06591, Republic of Korea

**Keywords:** urothelial carcinoma, renal pelvis, squamous cell carcinoma, liver, duodenum

## Abstract

Metastatic urothelial carcinoma of the renal pelvis (MUCP), a type of metastatic upper tract urothelial carcinoma (MUTUC), is a rare malignancy, and some patients with MUCP present with distant metastasis at the time of diagnosis. MUCP in the gastrointestinal tract is even rarer. Herein, we report a 78-year-old man with MUCP that presented as a duodenal ulcer. He complained of anorexia, dizziness, and melena for one month. Endoscopic examination at a local clinic revealed a duodenal hemorrhagic and ulcerative lesion, and the patient was referred. He noted dark-colored stools with increasing frequency, but he denied hematochezia, coffee ground emesis, weight changes, or abdominal pain. Gastroduodenoscopic examination at our hospital demonstrated an ulcerofungating lesion of the second portion of the duodenum. Colonoscopic findings showed no abnormality. Computed tomography showed a 6.7 cm sized mass abutting the inferior vena cava, second portion of the duodenum, lower pole of the right kidney, and right iliopsoas. The mass showed heterogeneous enhancement and internal hemorrhagic necrosis and infiltrated the perinephric soft tissues, the second portion of the duodenum, the right psoas muscle, the right renal vein, and the right adrenal gland. Duodenal biopsy showed moderately differentiated squamous cell carcinoma. Immunohistochemistry (IHC) showed diffuse and strong positivity for CK5/6. Tissue from the liver biopsy showed similar histopathologic features and showed GATA3 positivity on IHC. The imprint cytology smears of the liver tissue showed “cercariform” cell features. We confirmed the diagnosis as MUCP. This case illustrated a rare cause of a secondary duodenal tumor, MUCP.

Upper tract urothelial carcinoma (UTUC) is a rare malignancy that accounts for 5% to 10% of all urothelial cancers [1]. Advanced UTUC is aggressive and associated with a poor prognosis. At the time of diagnosis, 50−60% of patients with UTUC present locally advanced disease, and up to 25% present with distant metastasis [1,2,3]. Metastatic urothelial carcinoma of the renal pelvis (MUCP) comprises 67.5% of metastatic UTC (MUTUC). The most common metastatic sites are the regional lymph nodes, lungs, liver, and the bone system. MUCP in the gastrointestinal tract (GIT) is even rarer.

MUCP is rarely encountered in duodenal or liver biopsy samples that can closely mimic other malignancies. There is a wide cytological spectrum, ranging from a predominance of squamous cells in some cases, to a predominance of pleomorphic cells in others. According to a review article, cytologic diagnosis of MUCP has been rarely reported in the literature, despite extensive descriptions of the cytologic features of primary urothelial carcinoma (UC) from urine specimens [4,5,6].

We present a case of MUCP that presented as a duodenal ulcer and describe the characteristic histopathologic, cytologic, and immunohistochemical features.

A 78-year-old man complained of anorexia, dizziness, and melena for one month. Endoscopic examination at a local clinic revealed a duodenal hemorrhagic and ulcerative lesion, and the patient was referred. He noted dark-colored stools with increasing frequency, but he denied hematochezia, coffee ground emesis, weight changes, or abdominal pain. He has hypertension and takes clopidogrel. Laboratory investigations revealed mild anemia (hemoglobin of 8.0 g/dL), normal electrolytes, an elevated blood urea nitrogen of 32.4 mg/dL, and a creatinine level of 2.29 mg/dL, indicative of renal dysfunction and obstructive nephropathy. Gastroduodenoscopic examination at our hospital demonstrated an ulcerofungating lesion of the second portion of the duodenum (Figure 1A). Colonoscopic findings showed no abnormality. 

Unenhanced computed tomography showed a 6.7 cm sized mass infiltrating the inferior vena cava (IVC), the second portion of the duodenum, the lower pole of the right kidney, and the right iliopsoas (Figure 1B). Magnetic resonance imaging revealed a heterogeneously enhancing mass with necrosis in the lower pole of the right kidney involving the ureteropelvic junction and resulting in hydronephrosis (Figure 2A,B; asterisks). The aforementioned renal mass also infiltrated the perinephric soft tissues, the second portion of the duodenum (Du), the right psoas muscle (Ps), the ascending colon (Ac), and the IVC (Iv), combined with thrombosis in the IVC (Figure 2B,C). Two liver metastases were noted in the hepatic segment 7 (Figure 2D). Given its anatomic location, the duodenal mass was presumed to be renal cell carcinoma, duodenal cancer, or retroperitoneal sarcoma, such as leiomyosarcoma. 

Duodenal biopsy tissue showed moderately differentiated squamous cell carcinoma (SCC) (Figure 3A). Immunohistochemistry (IHC) showed diffuse and strong positivity for CK5/6 (Figure 3B). Tissue from the liver biopsy showed similar histopathologic features and showed GATA3 positivity on IHC (Figure 3C). The imprint cytology (IC) smears of the liver biopsy tissue were highly cellular, with densely cohesive clusters of epithelioid tumor cells (Figure 4A,B). The cells showed predominantly large and pleomorphic nuclei with moderate cytoplasm. Some cells showed unipolar cytoplasmic “tails” with flattened cells (Figure 4C,D; black arrows). The cytomorphologic findings from IC led to a preliminary differential diagnosis of poorly differentiated SCC or UC. Based on these histopathologic, cytologic, and IHC findings, the liver lesion was confirmed as MUCP.

Unfortunately, the patient underwent chemotherapy for one week, but his general condition decreased. The patient refused any further treatment and was transferred to a hospice care center.

MUCP is rare, and MUCP in GIT is even rarer. Diagnosing a MUCP in the GIT or liver poses a challenge [7]. Histopathologically, most renal pelvic and ureteral carcinomas are urothelial carcinomas (UC) with or without SCC components, but only 6% to 15% are classified as pure forms of SCCs. There is a wide cytological spectrum, ranging from a predominance of squamous cells in some cases of MUCP to a predominance of pleomorphic cells in others. Interestingly, the use of IC can be helpful as a valuable ancillary method because it can provide a more detailed cellular morphology than the histopathologic features. The following characteristic cell types commonly seen in metastatic UC have been described in the literature: giant and binucleate cells, fusiform and pear-shaped cells, plasmacytoid cells, and “cercariform” cells [8,9]. First designated by Powers and Elbadawi, “cercariform” cells are irregularly shaped cells with unipolar cytoplasmic processes with non-tapered, flattened ends [4,9]. In our case, cercariform cells showing unipolar cytoplasmic ‘‘tails’’ with flattened ends were observed on the IC slides (Figure 3C,D).

Ancillary immunohistochemistry (IHC) can assist in distinguishing UC from its potential mimics in cases of metastatic carcinoma of unknown primary origin [10,11]. Recently, GATA3 was highly recommended for diagnosing MUC, with high sensitivity and specificity in cell block from patients with MUC [11]. GATA3 expression also can be helpful in excluding carcinomas arising from the colon, pancreas, stomach, endometrium, ovary, and prostate. In our case, the IHC profile of the liver lesion was CK7+/CK20−/GATA3+, which supported the diagnosis of MUCP, and the profile of the duodenal ulcerative lesion was CK7−/CK20−/GATA3+.

Histopathologically, most UTUCs are UCs with or without SCC components, but 6% to 15% are pure forms of SCC [1]. The prognosis is equally poor for advanced stage (pT3 and pT4) MUCP. It is found that liver metastases and multiple organ metastases are related to poor survival in patients with MUCP [7]. Patients with MUCP stage T1–T2 tumors may be treated with radical surgery and show a good prognosis, while those with more advanced tumors often have metastatic disease. It is doubtful that there is a benefit of neoadjuvant or adjuvant radiotherapy/chemotherapy in the treatment of advanced MUCP [7]. Most patients treated with any of these modalities die of disease within months [12,13]. 

Diagnosing MUCP in the GIT or liver is challenging. Our case illustrates the characteristic cercariform cell features in the cytology, and the aid of immunohistochemical markers, such as CK7, CK20, and GATA3, contributes to establishing a correct and timely diagnosis.

## Figures and Tables

**Figure 1 diagnostics-13-02455-f001:**
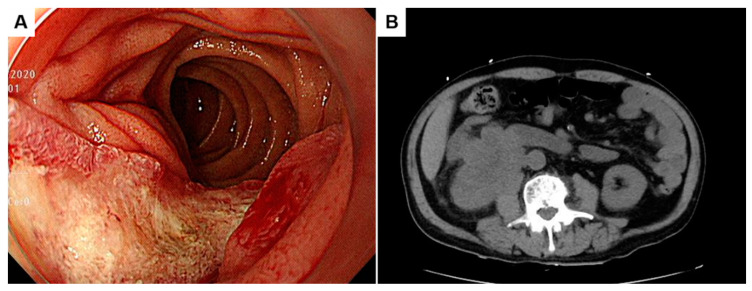
Gastroduodenoscopic finding of an ulcerofungating lesion of the second portion of the duodenum (**A**). Unenhanced computed tomography showing a 6.7 cm sized mass abutting the inferior vena cava, the second portion of the duodenum, the lower pole of the right kidney, and the right iliopsoas (**B**).

**Figure 2 diagnostics-13-02455-f002:**
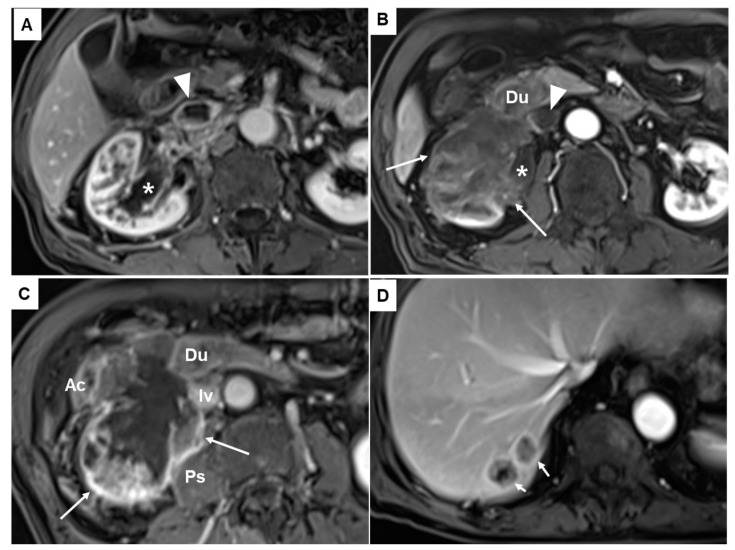
Fat suppression contrast enhanced T1-weighted magnetic resonance images revealed a heterogeneously enhancing mass (asterisks in (**A**,**B**)). The mass showed a necrotic portion (long arrows in (**B**,**C**)) in the lower pole of the right kidney infiltrating the pelvocalyceal system and ureteropelvic junction and resulting in hydronephrosis. It also invaded and infiltrated the perinephric soft tissues, the second portion of the duodenum (Du), the right psoas muscle (Ps), the ascending colon (Ac), and the inferior vena cava (IVC) (Iv), combined with thrombosis in the IVC (arrowheads in (**A**,**B**)). Two liver metastases were also noted in the hepatic segment 7 (short arrows in (**D**)).

**Figure 3 diagnostics-13-02455-f003:**
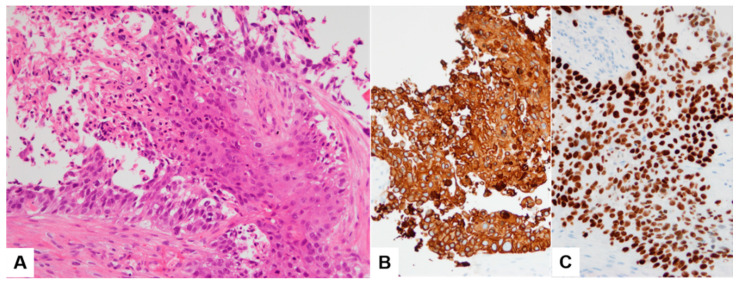
Histopathologic finding (**A**) and CK5/6 immunohistochemical stain (**B**) of duodenal biopsy tissue sample. Tissue from the liver biopsy showing GATA3 positivity on IHC (**C**).

**Figure 4 diagnostics-13-02455-f004:**
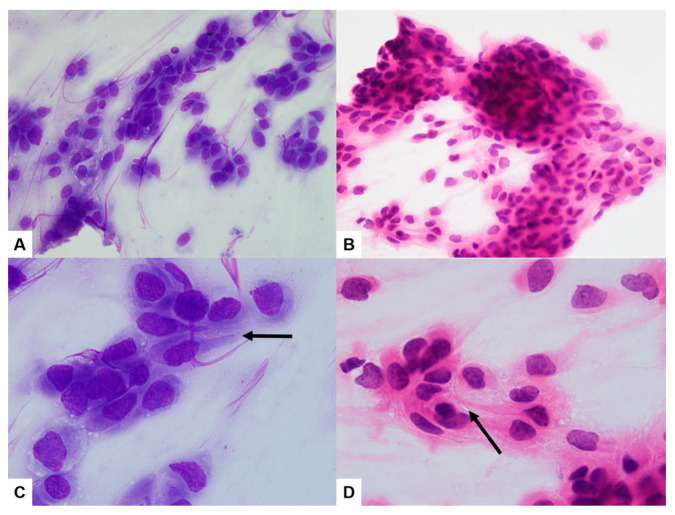
The imprint cytology smears of the liver biopsy sample were highly cellular, with densely cohesive clusters of epithelioid tumor cells ((**A**,**B**) ×400). The cells showed predominantly large and pleomorphic nuclei with moderate cytoplasm. Some cells showed unipolar cytoplasmic “tails” with flattened cells ((**C**,**D**) ×1000; black arrows) ((**A**,**C**) Diff-Quick; (**B**,**D**) H&E).

## Data Availability

The data presented in this study are available upon request from the corresponding author.

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
