# Peer review of "Unusual Duodenal Ulcer: Metastatic Urothelial Carcinoma of the Renal Pelvis"

_diagnostics, 2023, doi:10.3390/diagnostics13142455_

Round 1

Reviewer 1 Report

An interesting case report and with histopathologic description of Unusual metastatic urothelial carcinoma of the renal pelvis.

First, The case is not well described. Was there a follow up? Next, I would focus in the first paragraphs on the current evidence on rare metastatic sites of UTUC. Moreover, remove the first sentences on MUC of the Urinary bladder since this is misleading for a reader. The abstract should be modified as well. 

The paper should follow standard items of a scientific work: Introduction, mat and methods, case report and eventually a discussion.

UTUC is a rare disease and mUTUC is even less common and prognostic significance of the different sites is still unknown. Give a look at the following paper. https://doi.org/10.3390/jcm11185310

A narrative review of these rare metastatic sites can also be added in the discussion section.

The paper will be reconsidered after applying these suggestion.

Author Response

Dear Reviewer,

Thank you for your meticulous review of our manuscript. We truly appreciate your valuable comments. The manuscript has been revised and appropriate changes have been made based on your insightful and helpful comments. The following text represents point-by-point responses to the comments. Thank you again for your consideration and excellent feedback.

Your judicious comments have helped shape our manuscript into a better, more coherent version.

Point-by-Point Responses:

Comments and Suggestions for Authors

An interesting case report and with histopathologic description of Unusual metastatic urothelial carcinoma of the renal pelvis.

First, The case is not well described. Was there a follow up? Next, I would focus in the first paragraphs on the current evidence on rare metastatic sites of UTUC. Moreover, remove the first sentences on MUC of the Urinary bladder since this is misleading for a reader. The abstract should be modified as well.

Response: We agree with your comment that the first sentence on MUC of the urinary bladder is misleading readers. We deleted the first sentence and revised. We modified the abstract. We also described the clinical follow up.

The paper should follow standard items of a scientific work: Introduction, mat and methods, case report and eventually a discussion.

Response: We agree with your comment. We have revised the manuscript, and organized it in the order of introduction, method, case report, and discussion. We also added radiologic images and cytology microphotographs and added the findings in the methods part.

UTUC is a rare disease and mUTUC is even less common and prognostic significance of the different sites is still unknown. Give a look at the following paper. https://doi.org/10.3390/jcm11185310

A narrative review of these rare metastatic sites can also be added in the discussion section.

Response: Thank you very much for your valuable comment. We have cited the aforementioned reference and added a narrative review of these rare metastatic sites in the discussion section.

Reviewer 2 Report

The main advantages of this case report

-The article had a good level of English

-The authors have no conflicts of interest

The main disadvantages of this case report:

-Discussions and conclusions are missing.

-What tumor markers are useful?

-The blood test part is missing.

-There are no comorbidities?

-More bibliographic sources are needed

-What is the therapeutic attitude?

Author Response

Dear Reviewer,

Thank you for your meticulous review of our manuscript. We appreciate your valuable comments. The manuscript has been revised and appropriate changes have been made based on your insightful and helpful comments. The following text represents point-by-point responses to the comments. Your judicious comments have helped shape our manuscript into a better, more coherent version.

Thank you again for your consideration and excellent feedback.

Point-by-Point Responses:

Comments and Suggestions for Authors

The main advantages of this case report

 -The article had a good level of English

-The authors have no conflicts of interest

The main disadvantages of this case report:

-Discussions and conclusions are missing.

Response: We have revised and added discussion and conclusion.

-What tumor markers are useful?

Response: We have found that currently there is no useful tumor markers to detect upper tract urothelial carcinomas. Instead we have mentioned BUN and creatinine

-The blood test part is missing.

Response: We have added the blood test part and described anemia and renal dysfunction.

-There are no comorbidities?

Response: We have described comorbidities. He had hypertension and took clopidogrel.

-More bibliographic sources are needed

Response: We have cited more references on the upper tract urothelial carcinoma and metastatic urothelial carcinoma of the renal pelvis.

-What is the therapeutic attitude?

Response: We have described therapeutic options and prognosis.

Round 2

Reviewer 1 Report

The authors have addressed my main concerns. Suitable for publication in interesting imaging section

Reviewer 2 Report

I saw that the authors made the correct corrections. I agree with the publication in its current form